# Unveiling inter-embryo variability in spindle length over time: Towards quantitative phenotype analysis

**Yann Le Cunff**[¤], **Laurent Chesneau**, **Sylvain Pastezeur, Xavier Pinson, Nina Soler, Danielle Fairbrass, Benjamin Mercat, Ruddi Rodriguez-Garcia, Zahraa Alayan, Ahmed Abdouni, Gary de Neidhardt, Valentin Costes, Mélodie Anjubault, Hélène Bouvrais, Christophe Héligon, Jacques Pécréaux** *

CNRS, Univ Rennes, IGDR (Institut Genetics and Development of Rennes) – UMR 6290, Rennes, France

¤ Current address: Univ Rennes, CNRS, Inria, IRISA – UMR 6074, Rennes, France
* jacques.pecreaux@univ-rennes.fr

**Data Availability Statement:** A github repository (https://github.com/JacquesPecreaux/PCA_LeCunff_et_al_24) archived on zenodo (http://doi.org/10.5281/zenodo.11658220) offers a Jupyter-

## Abstract

How can inter-individual variability be quantified? Measuring many features per experiment raises the question of choosing them to recapitulate high-dimensional data. Tackling this challenge on spindle elongation phenotypes, we showed that only three typical elongation patterns describe spindle elongation in *C. elegans* one-cell embryo. These archetypes, automatically extracted from the experimental data using principal component analysis (PCA), accounted for more than 95% of inter-individual variability of more than 1600 experiments across more than 100 different conditions. The two first archetypes were related to spindle average length and anaphasic elongation rate. The third archetype, accounting for 6% of the variability, was novel and corresponded to a transient spindle shortening in late metaphase, reminiscent of kinetochore function-defect phenotypes. Importantly, these three archetypes were robust to the choice of the dataset and were found even considering only non-treated conditions. Thus, the inter-individual differences between genetically perturbed embryos have the same underlying nature as natural inter-individual differences between wild-type embryos, independently of the temperatures. We thus propose that beyond the apparent complexity of the spindle, only three independent mechanisms account for spindle elongation, weighted differently in the various conditions. Interestingly, the spindle-length archetypes covered both metaphase and anaphase, suggesting that spindle elongation in late metaphase is sufficient to predict the late anaphase length. We validated this idea using a machine-learning approach. Finally, given amounts of these three archetypes could represent a quantitative phenotype. To take advantage of this, we set out to predict interacting genes from a seed based on the PCA coefficients. We exemplified this firstly on the role of *tpxl-1* whose homolog *tpx2* is involved in spindle microtubule branching, secondly the mechanism regulating metaphase length, and thirdly the central spindle players which set the length at anaphase. We found novel interactors not in public databases but supported by recent experimental publications.

lab Python notebook to enable the reproduction of the computation of the PCA. It also features the data as comma-separated value files. To install this code, one will need an instance of conda; we used Anaconda (anaconda.org) and can install the proper environment using the provided yaml file PCA_code_published_from_history.yml.

**Funding:** RRG and JP were supported by a Centre National de la Recherche Scientifique (CNRS) ATIP starting grant and La Ligue nationale contre le cancer. HB was supported by EMBO through a long-term postdoctoral fellowship (ALTF 326-2013). We also acknowledge Plan Cancer grant BIO2013-02 to JP. JP's lab was supported by La Ligue contre le cancer. JP, HB and YLC were, each of them, supported by a Rennes métropole AIS grant. JP and HB acknowledge the support by Region Bretagne (SAD grants referenced AniDyn-MT and pRISM, respectively). The study was also supported by COST EU action BM1408 (GENiE). The funders had no role in study design, data collection and analysis, decision to publish, or preparation of the manuscript.

**Competing interests:** The authors have declared that no competing interests exist.

## Author summary

When quantifying the cell, scientists need to accurately measure cell-to-cell variability as it carries unique information about the underlying mechanisms. In this article, we focused on the spindle length as a proxy for correct mitosis. We used the nematode one-cell embryo, an established model organism, to investigate (stem-like) cell divisions. Through a data-only approach of spindle dynamics over time, we automatically extracted the most informative variability descriptors. We recalled two known ones: spindle length and anaphase elongation rate. We uncovered a new one –late-metaphase shortening –present in all conditions. Such a phenotype was previously confined to cells with defective chromosome attachments. These three descriptors account for 95% of variability, suggesting that the complex spindle choreography relies only on a few core mechanisms. Furthermore, we showed that the final spindle length at anaphase, important to set the daughter cell fates, is already determined in late metaphase, despite a complete spindle reassembly between both phases. Interestingly, the same descriptors explain variability in genetically perturbed and non-treated conditions. This suggests that no novel mechanism appears in defective cells. Only mechanism contributions are changing. Finally, we propose a tool to predict genes co-involved in a mechanism from a known gene to support candidate approaches.

## Introduction

While cell division is remarkably faithful, the mitotic spindle, key to ensuring a correct partitioning of the chromosomes, can take variable trajectories to achieve its task, varying its shape, organisation or overall position in the cell [1–3]. Phenotype variation between genetically identical cells can arise from multiple causes, from random transcription rates to stochastic variations in internal chemical reactions [4–6]. This emergent variability can translate either into a gradation of phenotypes across cells, e.g. in expression levels of a fluorescent dye [7], or radically new cell behaviours through threshold effects, e.g. responsiveness to the induction of apoptosis [8]. Along that line, Raj et al. [9] investigated the development of the *C. elegans* intestine: high fluctuations in gene expression can lead to either viable or impaired intestinal development, even in a clonal population [9]. Unlike this latter study, which uncovers the variability mechanism, it is generally challenging to identify whether a perturbation leads to quantitative or qualitative changes in phenotype. That is, whether one observes diverse grades of the same phenotypes or the appearance/disappearance of a novel phenotype. While the most common interpretation of phenotype variation between identically genetically-perturbed cells is differences in protein quantities, it may also reveal a loosely constrained system [10–13]. Thus, variability in spindle trajectories was often viewed as noise, although it fosters the spindle's ability to resist or adapt to internal defects like chromosome misattachment and external perturbations like changes in tissue environment [14–20]. Consistently, variability may increase cellular fitness in cancer [21]. Our paper proposes a methodology to characterise the nature of variability (qualitative versus quantitative) with a thorough quantitative analysis, using the well-studied and stereotypical cell division of *C. elegans* one-cell embryo as a representative example [22].

To quantify cell-to-cell variability, one can choose a specific quantitative feature and display the variance over a given population, e.g. size of internal structures such as spindle or centrosomes [23] or expression of a given gene of interest [7]. Studying variability in dynamical

systems, like spindle kinematics during cell division, raises subsequent technical difficulties, namely measuring variability between trajectories. Disregarding whether one focuses on mechanics or biochemistry, mathematical modelling can be instrumental [24–26]. The issue then shifts from comparing trajectories to comparing parameters from the fitting of the mathematical model to the experimental data, *id est* measuring the variance of each model parameter across a population of cells to capture the overall cell-cell variability. Although attractive, such an approach requires an established and integrative model, based on a priori assumptions, to recapitulate all the phenotypes observed experimentally. In contrast, we rather adopted a data-centred perspective without modelling assumptions to uncover characteristics revealing still unknown mechanisms. Similarly, we also avoided manually selected features.

Numerous models describe how wild-type cell divisions occur. However, because of the complexity resulting from both numbers and families of molecular actors involved, it is often required to focus on one part of cell division [27–30]. In particular, the correct partitioning of the chromosomes critically depends on the metaphasic spindle. Its functioning, especially the role of mitotic motors and microtubules, received much attention [16, 31–33]. The spindle length is a classic and suitable entry point [34, 35]. It was already investigated through biological or modelling means [36–39]. However, its mere observation left open many potential mechanisms, reflected in a broad range of non-consensual models, none recapitulating its dynamics fully. That is, tractable models usually rely on simplifying assumptions to capture the core principles of the biological system they represent, overlooking sometimes specific cell-cell differences. In contrast to these reductionist approaches, our data-centred approach, not relying on any hypothesis, enabled us to discover novel and unexpected behaviours of the spindle, although it left open the question of their mechanistic explanations.

In this paper, we classified spindle-elongation cell-cell variability using dimensionality reduction algorithms, a common paradigm in biology [40]. From the spindle length over time, measured during mitosis of the *C. elegans* one-cell embryo, we extracted Principal Component Analysis (PCA) projection as a blueprint. We investigated how each embryo may depart from the average or control blueprint in genetically perturbed conditions to gain a phenotype taxonomy. We aimed to measure the variability observed across different embryos beyond the usual all-or-nothing classification and derived interpretable archetypes of such variability, aiming towards quantitative phenotypes. To achieve this program without a priori, we focused on data projection methods that reduce dimensionality, the most famous being PCA [41]. We expected the PCA to extract cell division's key features and highlight biologically relevant mechanisms while discarding noise. Such an approach has been used to classify cell phenotypes according to gene expression or cell shapes [42, 43]. At the organism scale, it enabled describing nematode shapes and movements quantitatively across various strains and provided a list of descriptors to do so [44]. We set to apply a similar approach to spindle dynamics during the division of the *C. elegans* one-cell embryo. As we study trajectories over time, our automatically derived descriptors of spindle elongation (further named archetypes) can be seen as typical trajectories. Experimental elongations are then a linear combination of these archetypes added to the average elongation. Our hypotheses-free archetypes partly meet the features largely adopted by the community, like the average spindle length and elongation rate. Interestingly, it also uncovered the novel and unexpected transient spindle shortening archetype, which, combined with others, limited late-metaphase elongation.

## Results

To understand the grounds of spindle behaviour variability, we derived archetypes of variability without a priori using PCA. Our dataset included spindle elongation sampled at typically

30 frames per second, from 1618 experiments. We targeted 78 genes, 52 with cell-division-variant phenotype or one of its descendants in the worm phenotype ontology [27, 45, 46], out of 1191. We included all microtubule-binding molecular motors (Methods). We assumed that they were enough to challenge our approach as the dataset was diverse enough while requiring a reasonable dataset-acquisition effort to record embryo movies suitable for this study. Indeed, acquisitions had to be manually performed to reach the required accuracy.

## Cell-cell variability in spindle elongation is recapitulated into three archetypes

**Diversity of spindle elongation phenotypes.**   After filtering out the centrosome-tracking outliers (Methods), we computed the spindle length as the distance between the two centrosomes (Fig 1A–1C). We observed a variety of phenotypes departing from the average behaviour of the whole dataset, even restricting observations to embryos from the same strain and at an identical temperature (Fig 1D). Interestingly, a diversity of phenotypes was also visible among the non-treated condition alone, as exemplified by GFP::ɣTUB strain (TH27) embryos in Fig 1E. The average spindle elongation for non-treated embryos (blue dashed curve) displayed two-step dynamics during metaphase and anaphase: A first mild increase in spindle length started about one hundred seconds before anaphase onset (referred hereafter as metaphase elongation), followed by a second quicker increase (referred hereafter as anaphase elongation), corresponding to the elongation in the literature. These two-step dynamics were observed in both the average elongation over the whole dataset and wild-type embryos. It is reminiscent of the "biphasic metaphase" observed by Goshima and Scholey [34]: in unperturbed condition, a quite constant spindle length in early metaphase followed by a rapid increase in late metaphase. Interestingly, the timing of the first elongation matches the one of spindle-shortening before anaphase onset upon defects in kinetochore–microtubule attachments [47–49]. For instance, targeting CLS-2, a CLASP protein stabilising microtubules particularly at the kinetochore [50–52], we observed such a spindle shortening meanwhile the anaphase elongation also departed from the control to a variable extent. The three upper trajectories in Fig 1F (identified with an arrowhead) were typical of the so-called "spindle weakening" phenotype, resulting in a faster elongation of the spindle than the control one. We arbitrarily assumed that fast elongation corresponds to anaphase onset in this condition. In a broader take, protein depletions caused a variety of phenotypes, raising the question of how to describe such a wide inter-individual variability.

**Using projection methods to obtain a model-free description of variability.**   We set out to achieve interpretable, global, and without a priori classification and quantification of the spindle elongation phenotype. To ensure phenotype interpretability, we used an unsupervised linear classification so that phenotypes could be seen as the superposition of biologically meaningful archetypes (see below). Our approach contrasts with deep-learning-based approaches, which are attractive because they take the images directly as input. But these methods are often intended for classification only, for instance, in anatomopathological work, rather than fundamental research [53–55]. Because a substantial fraction of the components involved in mitosis mechanics are involved both in metaphase and anaphase, we set out to classify elongation curves without splitting them into these two phases to investigate an interdependence of the phenotypes between phases. It contrasts with the classic view of separating the modelling because distinct structures are assembled in metaphase and anaphase [37, 56, 57]. This was technically reflected in avoiding local projection methods. Finally, we also aimed to perform an analysis without a priori, so we focused on feature extraction and unsupervised algorithms [58]. The eigenvectors of our projection method, named here archetypes, played the role of

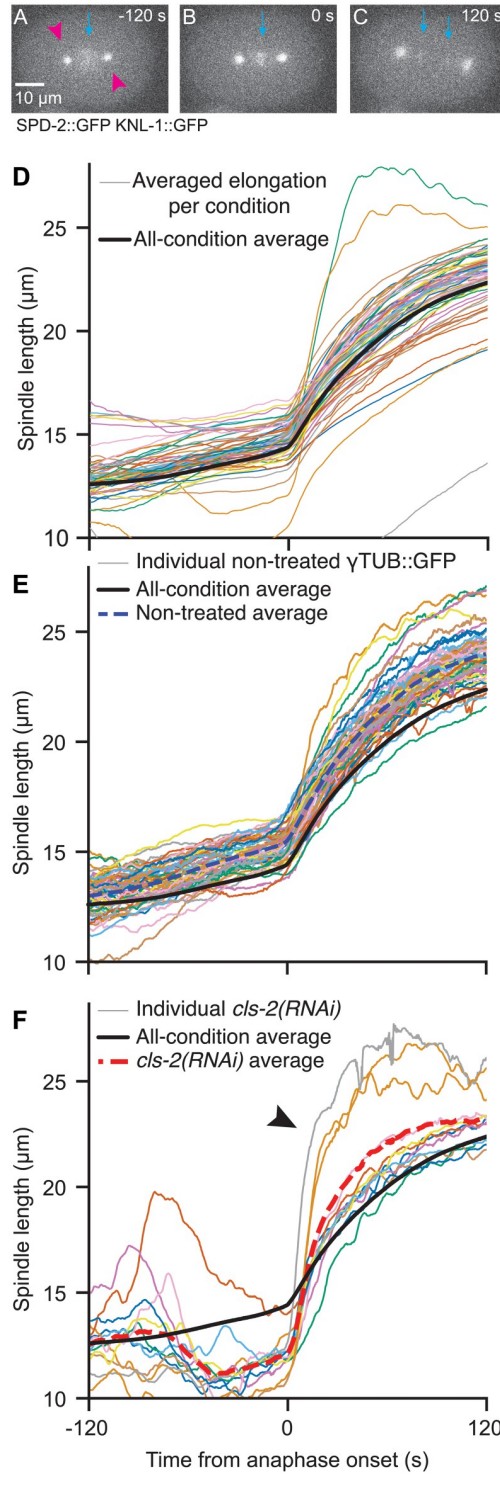

**Fig 1. Diversity of the spindle length phenotypes at 18°C.** (**A-C**) Exemplar stills of a typical embryo of strain JEP29, 120 s before, at anaphase onset and 120 s after. (magenta arrowheads) Centrosomes were labelled through SPD-2::GFP and (cyan arrow) kinetochores through KNL-1::GFP. The scale bar represents 10 μm. (**D**) (thin coloured lines) Pole-pole distance (spindle length) averaged per condition and plotted during metaphase and anaphase for each of the 67 conditions, including 3 non-treated distinct strains, one control, and 63 gene depletions/mutations. All included embryos were imaged at 18°C (S1 Table). Conditions with less than 6 embryos were not represented. Multiple conditions treating the same gene by RNAi or mutating it are merged. (**E**) The thin coloured lines depict the spindle lengths for individual non-treated TH27 embryos (*N* = 58), and the thick blue dashed line is the average of all embryos

of this condition. (**F**) (thin lines) Spindle length for individual *cls-2(RNAi)* treated embryos (N = 12). The thick red dashed line corresponds to the average spindle length over these embryos. Arrowhead indicates the tracks showing fast spindle elongation revealing "spindle weakening". In panels D-F, individual embryos and averaged tracks were smoothed using a 1.5 s-running-window median. The black thicker line corresponds to the average over the whole dataset, including all conditions.

features but are directly extracted from the data. In this, our approach contrasts with the previous classification of mitosis phenotype using manually determined features [2, 23]. Indeed, we aimed to improve the variance explicability and keep the opportunity to discover novel behaviour, paving the way towards uncovering unexpected behaviours and thus mechanisms. Interestingly, and similar to the manual-features-based method, the dosage of archetypes offers a quantitative view of the phenotypes.

We benchmarked several approaches and chose the one that maximised the within-condition to between-condition variability ratio (S1 Methods). The Principal Component Analysis (PCA) turned out to yield the second-best score (S4 Table), while local linear embedding gave the best one. However, we used PCA as it was global and interpretable. Together with projecting trajectories, PCA produced eigenvectors, further termed archetypes, and allowed to compute their dosages in each embryo spindle-length curve, named below coefficients. Surprisingly, the first three archetypes of the PCA were necessary but also sufficient to provide most of the information needed to describe variability in our dataset, about 95% (S1(B) Fig). We further used three components, as it appeared to be the corner value of the L-curve, in an analogy to finding an optimum number of clusters in k-means algorithm or hierarchical clustering (S1(A) Fig) [59]. For each elongation curve, the timepoint-wise difference from the whole-dataset average (Fig 1D, thick black curve) could be recapitulated into the weight given to each archetype (coefficient), i.e. the dosage of each of the three archetypes. Importantly, these latter were reminiscent of elongation patterns found in some perturbed conditions, either using RNAi or mutant (Fig 2A), suggesting they could have a biological interpretation. An exemplar reconstruction illustrates how three components were enough to account for a given single-cell spindle elongation (S1(C) Fig).

**Three main archetypes of spindle elongation.** These main archetypes accurately describe the diversity in spindle elongation. The first archetype explained 70% of the overall variability in spindle elongation trajectories (Fig 2A, blue curve, and S1(B) Fig). Its roughly flat dynamics suggested that it accounted for the average spindle length, a classic feature in studying the spindle dynamics [2, 23]. A higher corresponding coefficient reflects an upshifted spindle-length curve (Fig 2B). The second archetype explained 19% of the variability (Fig 2A, red curve, and S1(B) Fig) and comprised two elongation phases, during the metaphase (-100 to 0 s from anaphase onset) and the anaphase (0 to 50 s). Therefore, the second archetype mainly captured the dynamics of spindle elongation. A high corresponding coefficient reflected a fast elongating spindle (Fig 2C). It is also a classically used feature, although only at anaphase [2, 23]. The third archetype explained 6% of the variability (Fig 2A, green curve, and S1(B) Fig) and corresponded to at least an inflexion in the spindle elongation up to a plateauing that limited late-metaphase elongation and is observed in non-treated embryos, e.g. (Fig 2D and S2 Fig) [60]. When more weighted, this archetype accounted for a spindle shortening, especially in *cls-2 (RNAi)* (Fig 1F). Importantly, this third archetype found nothing comparable in the previous studies, highlighting the interest of our data-centred approach. Beyond proposing archetypes, we found that only three components (archetypes) are needed to recapitulate most of the spindle elongation kinematics, suggesting that few mechanisms underlie the complexity of spindle elongation phenotypes.

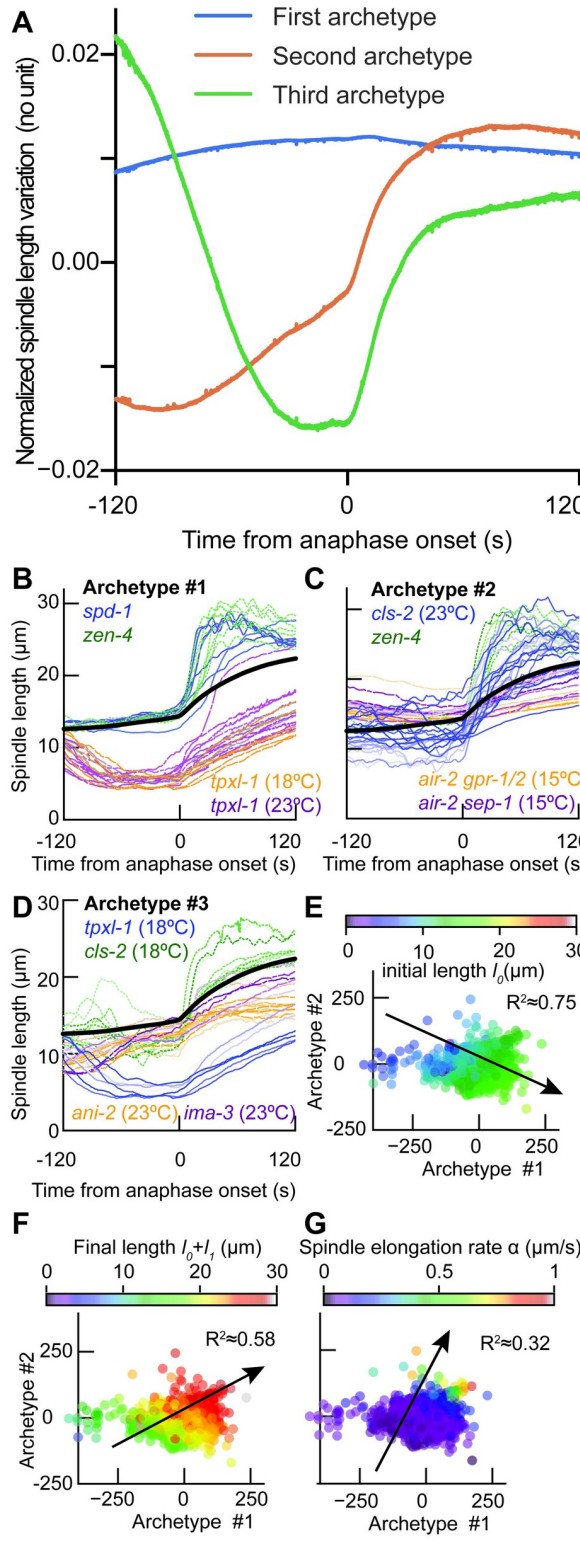

**Fig 2. Principal component analysis of the spindle elongation and corresponding archetypes (eigenvectors).** (**A**) The three main archetypes extracted by PCA over all conditions account for more than 95% of cell-cell variability in spindle elongation (see main text for their plausible interpretation). (**B**) Spindle elongations from individual experiments for the two conditions resulting in the highest coefficients for archetype 1 averaged over the condition ($\overline{C_1}$), namely GFP::γTUB (TH27) embryos treated with *spd-1(RNAi)* (N=7, $\overline{C_1} \simeq 96.6$) and *zen-4(RNAi)* (N=7,

$\overline{C_1} \simeq 116$). The lowest coefficients for archetype 1 were obtained by treating the same strain with *tpxl-1(RNAi)* and imaging at 23°C ($N$=12, $\overline{C_1} \simeq -327$) or 18°C ($N$=7, $\overline{C_1} \simeq -366$). (**C**) Similar assay for archetype 2. Highest values were obtained with GFP::γTUB (TH27) embryos treated with *cls-2(RNAi)*, imaged at 23°C ($N$=21, $\overline{C_2} \simeq 81.2$) and *zen-4 (RNAi)* imaged at 18°C ($N$=7, $\overline{C_2} \simeq 101$). The lowest coefficients for archetype 2 were obtained treating the *air-2 (or207)* mutant labelled with KNL-1::GFP and SPD-2::GFP (JEP31 strain) either with *sep-1(RNAi)* ($N$=11, $\overline{C_2} \simeq -95.2$) or *gpr-1/2(RNAi)* ($N$=9, $\overline{C_2} \simeq -114$), and imaging at 15°C. (**D**) Similar assay for archetype 3. Highest values were obtained with GFP::γTUB (TH27) embryos treated with *tpxl-1(RNAi)* ($N$=7, $\overline{C_3} \simeq 62.2$) and *cls-2(RNAi)* ($N$=12, $\overline{C_3} \simeq 50.6$) while lowest coefficients resulted from treating with *ima-3(RNAi)* ($N$=5, $\overline{C_3} \simeq -41.2$) and *ani-2 (RNAi)* ($N$=13, $\overline{C_3} \simeq -48.2$). All considered embryos in panels (B-D) were imaged at 18°C except otherwise stated. (**E-G**) Fit of spindle elongation curves after [23] and mapping the corresponding parameters on the PCA plane (see main text and S1 Text): (**E**) Average spindle length in early metaphase $l_0$ and (**F**) in anaphase $l_0 + l_1$, and (**G**) elongation rate $\alpha$. These three values are colour-coded, and an arrow depicts the axis along which these quantities vary (gradient) while we plotted the values of the two first coefficients of the PCA for embryos from the whole dataset except 16/1618 embryos whose elongation cannot be fitted. 14 additional fits were excluded because of aberrant values. The adjusted R-squared is reported for the multilinear fits, and the corresponding p-values are below $10^{-15}$) (S1 Methods).

**Comparing with existing spindle characteristics.** Having proposed three hypothesis-free archetypes to characterise the spindle, we next asked how they compared to the previously educated-guessed ones. In the nematode, to study the genetic grounds of cell-division variability, Farhadifar and colleagues pre-computed about 20 features, such as embryo size, division duration or centrosome oscillation duration and frequency, before applying a PCA to have a quantitative genetics approach to nematodes spindle across strains and species [23, 23]. Through this comparison, we will also gain a better understanding of the corresponding biological mechanisms to our archetypes. We thus fitted individual embryo elongation curves with Farhadifar's model, using non-linear least-squares to extract three of their features: the average spindle length at early metaphase, the final spindle length in anaphase, and the elongation rate (S1 Text). We then mapped the above manual features on the two first PCA dimensions (Fig 2E–2G and S1 Methods). We found that the initial length is mostly related to our first coefficient, as depicted by the black arrow (Fig 2E). Meanwhile, the spindle's final length combined coefficients 1 and 2 (Fig 2E). Finally, we found a good alignment of the elongation rate with our second coefficient of the PCA, as expected (Fig 2G). Our third coefficient appeared as a novel feature to characterise the spindle elongation that did not correlate with Farhadifar's features (S1 Text). It demonstrated the ability of our data-centred approach to uncover original features, paving the way towards understanding novel mechanisms, meanwhile, our two first coefficients departed somewhat from pre-existing features, as suggested by the non-horizontal/vertical black arrows (Fig 2E and 2F).

## Robustness of the main archetypes

**Robustness to dataset composition.** The three archetypes are automatically extracted from the dataset to describe its heterogeneity best. Therefore, one can wonder whether the rather extreme phenotypes, corresponding to high coefficients, are defining the archetypes (Fig 2B–2D). In other words, how robust are these archetypes with respect to the composition of the dataset? To investigate this issue, we performed a bootstrap sampling of the experimental dataset. We randomly selected 500 experiments among the 1618 present in the dataset and repeatedly computed the archetypes (eigenvectors of the PCA) (§4 in S1 Methods). We found mild changes in archetypes despite considering only 31% of the dataset (Fig 3A). It suggests that our archetypes are present in most of our embryos rather than set by a few extreme phenotypes.

Going further, we focused on the non-treated embryos. These embryos demonstrated no extreme phenotype with a spindle elongation close to the average behaviour, that is, the (0,0,0) position in the PCA space (as shown in S3 Fig). Strikingly, we found similar archetypes (Fig 3B). Reciprocally, we tested the use of only the treated embryos (excluding the control experiments using L4440 RNAi) and again found similar archetypes (Fig 3C). Furthermore, the contributions of each archetype to describe the overall variability of non-treated embryos are comparable to those obtained with the whole dataset (S5 Table). We concluded that mild phenotypes, control and non-treated conditions are well represented by our proposed three archetypes.

Genetically perturbed embryos, with some extreme phenotypes, relied a bit more on archetype 2, mildly increasing their corresponding explained variance. Finally, and as a control, we selected the embryos from the functional group *kt* as they display a very peculiar elongation pattern (S1 and S5 Tables and S6(B) Fig). Not surprisingly, PCA applied to these embryos resulted in different archetypes (S6 Fig). This test acted as a positive control, challenging the dependence of archetypes on data, suggesting that a diverse enough set of data is required to well extract the features.

**Robustness to experimental conditions.** Variability can also arise from changes in the environment to which the organisms might respond. Worms in their natural environment

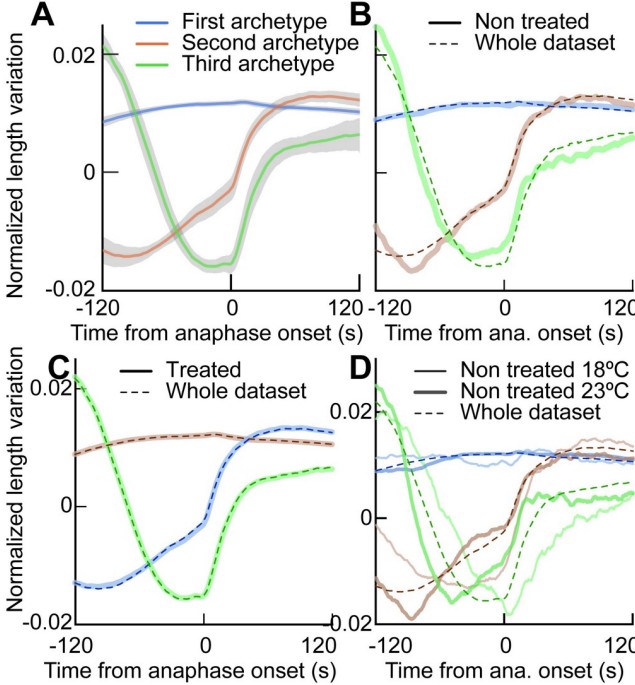

**Fig 3. Variations of the archetypes upon dataset changes.** (**A**) (lines) Average of the three first PCA archetypes (eigenvectors) ±(shading) two times their standard deviations, computed over a 500 embryos subset of the data (disregarding conditions). We repeated this bootstrapping 500 times to obtain standard deviations. (**B**) (thick lines) Three first PCA archetypes were computed considering only the non-treated conditions (*N*=129) and compared to (dashed lines) archetypes extracted from the whole set of conditions (*N*=1618). (**C**) (thick lines) Three first PCA archetypes computed considering only the treated conditions (*N*=1308), meaning RNAi or mutant without L4440 controls, and compared to (dashed lines) archetypes extracted from the whole set of conditions. **D**) Three first PCA archetypes computed considering only the non-treated conditions at (thick lines) 23°C (*N*=71) and (thin lines) 18°C (*N*=58), compared to (dashed lines) archetypes extracted from the whole set of conditions. In all panels, the elongation curves were smoothed with a 1.5 s running-median filtering before computing PCA. Explained variances are reported in S5 Table.

experience various kinds of stress, among which changes in the temperature have received attention [61, 62]. We wondered whether variability in spindle elongation would have the same underlying archetypes at two different temperatures since they correspond to two different spindle elongation phenotypes (S4 Fig). We repeated the PCA projection considering two sets of non-treated embryos at 18°C and 23°C, and observed that the general shape of the archetypes are similar across temperatures and compared to the ones extracted from the whole dataset (Fig 3D). Importantly, as expected, the archetypes appeared horizontally squeezed at the higher temperature because the division pace is increased [62]. It likely reflects a higher dynamics of molecular components, like the microtubules and molecular motors, although all are not identically sensitive to temperature changes [63–65]; for instance, it leads to faster anaphase spindle rocking at higher temperature [61]. It was also noteworthy that the second archetype did display a first elongation in late metaphase, not visible in the 18°C counterpart.

Overall, we concluded that provided that a variety of elongation patterns was fed into the PCA through either a large enough set of non-treated embryos or a variegated set of protein depletions, the archetypes' general shapes were conserved across temperatures, a panel of genetically perturbed embryos and even non-treated or control ones. It suggested that while dynamics of elongation are influenced by these changes, the nature of variability did not change. Therefore, our data-centred archetypes are generic enough to be instrumental in analysing a broad range of phenotypes.

## Anaphase onset as a turning point

At metaphase and anaphase, distinct structures, namely the metaphasic spindle and the central spindle, connect the spindle poles. Consistently, manually set features focus on a single phase. In contrast, our two first archetypes spanned metaphase and anaphase and accounted for 89% of the variability. It calls for investigating the interdependence of the spindle between both phases. We computed the spindle length in late anaphase ($l_{LA}$) as the average of the 300 last data points spanning between 111.7 s and 120 s after anaphase onset. It can be viewed as a marker of correct spindle functioning, but beyond, correlates with spindle final position and, therefore, cytokinetic furrow positioning [22, 66–69]. We used a machine learning approach to investigate whether late anaphase spindle length was set early during division (§3 in S1 Methods). Spindle length during a 25 s interval starting 5 s before anaphase onset provided an accurate prediction of length minutes later (purple and green shadings, respectively, Fig 4A). Comparing predicted spindle late-anaphase lengths and experimental ones, we obtained a Pearson coefficient $R = 0.82$ with $p < 10^{-15}$ (Fig 4B). We next wondered how this correlation depended on the starting time of the 25 s interval. Considering Pearson R as the *predictive power*, we computed it when sliding the interval (Fig 4C). We observed a steep increase in the predictive power around anaphase onset. In other words, most late-anaphase spindle length variability depends on the mechanisms already present at anaphase onset.

We next asked whether this information announcing the late-anaphase spindle length was captured by PCA projection. We thus trimmed the elongation curves to the (-5, 20) s interval, did the PCA projection as described above and kept only three components (archetypes). We then performed a similar machine learning assay (§3 in S1 Methods). We obtained a Pearson coefficient $R = 0.80$ with $p < 10^{-15}$ (Fig 4D) when comparing predicted and measured late-anaphase lengths. We concluded that the spindle characteristics supporting the prediction of the late-anaphase length were well recapitulated by the coefficients of the three archetypes, at least in late metaphase and early anaphase.

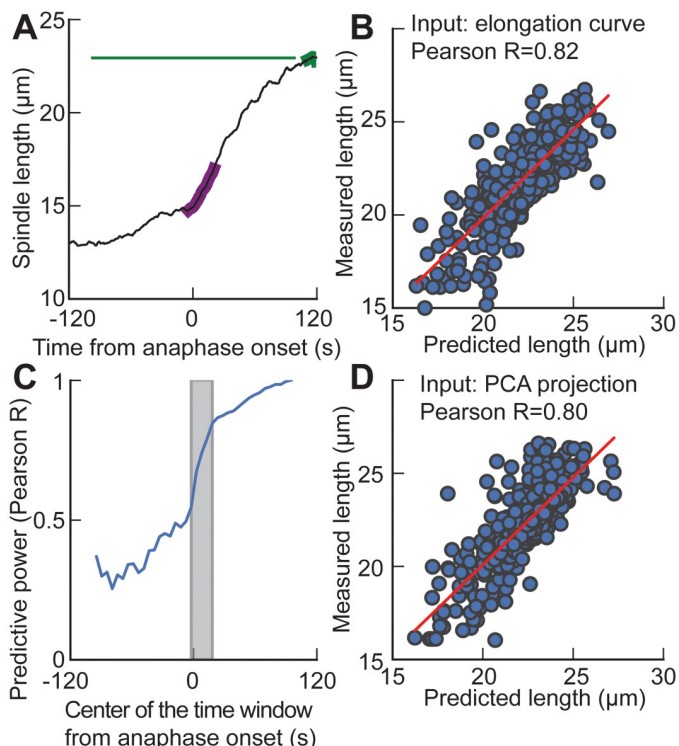

**Fig 4. Machine learning (ML) predicts the spindle late-anaphase length $l_{LA}$.** (**A**) (black line) Elongation of an exemplar non-treated GFP::γTUB embryo, imaged at 18°C, highlighting (purple) the $(-5, 20)$ s interval used as input of algorithm and (green) the $(111.7, 120)$ s interval from anaphase-onset to compute the average spindle length at late-anaphase. (**B**) Using embryos in the testing set (32%, i.e. 576), we plotted the measured spindle late-anaphase length versus the predicted final length and obtained a high correlation ($R = 0.82, p < 10^{-15}$) when inputting an elongation curve over the (-5, 20) s interval as input to the ML network. (**C**) We slid this 25 s interval by 5 s starting at $-120$ s and ending at 120 s, and computed the Pearson coefficient, as above, denoted predictive power. The grey shading region corresponds to the interval used in other panels. (**D**) Using embryos in the testing set, we plotted the measured spindle late-anaphase length versus the predicted final length and obtained a high correlation ($R = 0.80, p < 10^{-15}$) when inputting the PCA projection (coefficient) computed on the elongation curve over the (-5, 20) s interval as input to the ML network. Details about the algorithmic approach are provided in §3 in S1 Methods.

## From archetypes to phenotypes

**Interplay between the spindle elongation phenotype and the projection.** We used partial RNAi-mediated protein depletion in many cases (S1 Table). We reckoned that the shift of PCA coefficients from control may depend on the penetrance of the RNAi. Providing a general demonstration of such a link would be out of the scope of this paper. We instead offered an example: we varied the amount of a known microtubule-associated protein, the depolymerising kinesin KLP-7. Its depletion caused decreased microtubule growth and shrinkage rates, and reduced rescue and catastrophe rates [51, 70], causing microtubule-chromosome attachment defects [71, 72]. It is reported to cause only mild spindle length defect upon hypomorphic treatment (Fig 5A), while spindle breakage happens only upon penetrant one [73, 74]. It made KLP-7 an excellent candidate to test the link between penetrance and PCA coefficients. Thus, we investigated embryos where this protein was fused with mNeonGreen at the locus as a protein level reporter and depleted by RNAi. We observed a larger first coefficient in correlation with a higher amount of fluorescence, i.e. depletion causes a lower "dose" of archetype 1 (Fig 5B). It corresponded to a Spearman correlation coefficient between the first coefficient and the fluorescence of $\rho = 0.35$ ($p = 0.016, N = 36$). Because of the low number of

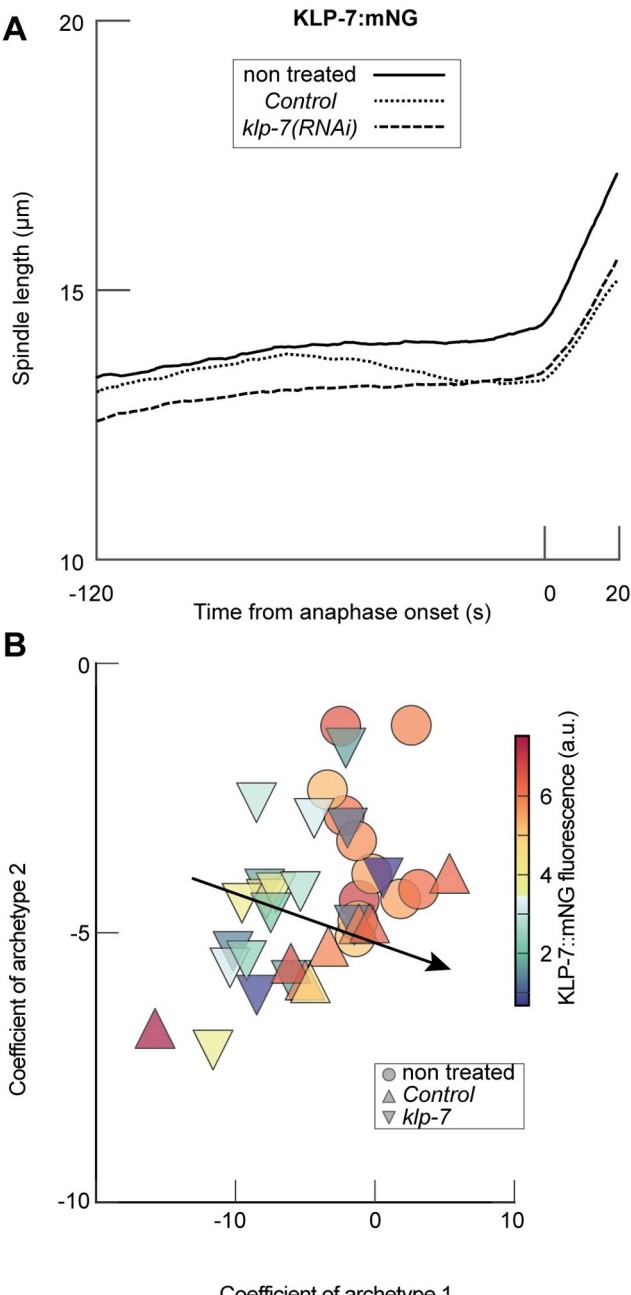

**Fig 5. PCA coefficients depend on the penetrance of RNAi. (A)** Average spindle elongation trajectories for the depletion of KLP-7$^{MCAK}$ in KLP-7::mNG background (strain LP447) in three conditions: (dotted lines) *N*=18 *klp-7 (RNAi))* treated embryos; (dashed lines) *N*=8 control embryos (L4440 treated); and (plain lines) *N*=11 non-treated embryos. Individual embryos elongation are reported in S8 Fig. **(B)** Projection of these conditions on the PCA established on the whole dataset. We plotted the coefficients corresponding to the two first archetypes. The black arrow depicts the axis along which this fluorescence varies (gradient), computed similarly to Fig 2E, 2F and 2G. The three conditions reported here were not included in the initial dataset used to generate PCA archetypes. Acquisitions were performed at 18˚C. The marker colour encodes the fluorescence level of KLP-7::mNG (Methods).

samples, we used a permutation test to show a mild significance of this correlation. The second and third coefficients did not appear significantly correlated with the level of fluorescence. In the case of KLP-7, the variability of the phenotype, quantified by our PCA coefficients, reflects the grading of depleting by RNAi. Conversely, it suggests that variability of penetrance likely results in spreading the coefficient values within a given condition cloud. Other causes of spreading are likely present. Indeed, the non-treated condition also inherently displays variability (S3 Fig and S2 File) corresponding to the distribution of elongation curves (Fig 1E).

**From spindle elongation pattern to gene function.** Having established that the set of PCA coefficients corresponding to a condition is a robust and *bona fide* representation of its phenotype of elongation, we reckoned that the median position over all replicas of the same condition may map the phenotype in the PCA plane by linking gene functions and the weights of the archetypes. We computed the median coefficient for each condition and mapped them in the PCA plane. Since gene ontology terms or worm phenotype ontology turned out to be too generic, we tagged each condition manually with the main function of its corresponding protein (group column in S1 Table) during embryonic division. Plotting the corresponding groups in the PCA coefficient space indicated partially overlapping clusters (Fig 6 and S1 File). To test to which extent each coefficient supported this grouping, we used the Kruskal-Wallis test for each coefficient. While coefficient 1 did not really support clustering ($p = 0.12$), coefficients 2 and 3 are discriminative ($p = 0.0044$ and $p = 7.34 \times 10^{-5}$, respectively). We shuffled the group labels and computed the Kruskal-Wallis $H$ values; we repeated this 10000 times, and obtained the corresponding distribution (S5 Fig). We concluded that our PCA analysis could be instrumental in suggesting gene function by looking at their neighbours in the PCA space.

**The PCA's three coefficients are a predictive tool.** To illustrate the ability of the archetype's coefficients to predict gene-phenotype association, we first investigated TPX2$^{\text{Tpxl-1}}$ as its role in the nematode zygote spindle is surprising, not being reported as involved in spindle-microtubules branching, while branched microtubules are observed by cryo-electron microscopy [75]. Furthermore, no homolog of the Augmin/HAUS complex was found, and the C. *elegans* γ-TURC differs also [76]. In contrast, its Xenopus and mammalian homologs were proposed to contribute to creating a branched microtubule network in the spindle with the help of XMAP215$^{\text{Zyg-9}}$ [77, 78]. To find genes functionally related to *tpxl-1* in the nematode, we looked at conditions with similar PCA coefficients. Beyond visual inspection of the map (S1 File), we sought a method to determine related genes. Considering *tpxl-1* as the gene of interest, we listed the interactions known or predicted in wormbase [79] and whose corresponding genes were present in our dataset. We considered that there is no interaction with L4440 and non-treated conditions. From these known conditions, we used machine learning and trained a logistic regression fed by the three PCA coefficients. We used it to predict whether the other genes in our dataset are interacting (S1 Methods). For *tpxl-1*, we found 14 predicted interactions not yet present in wormbase among 16 interactions overall (S7 Table). Among these, we did not find *zyg-9*, consistent with a branching mechanism departing from the one documented in Xenopus or mammals [77, 78]. Furthermore, we did not find *plk-1* differing from the proposed nucleation mechanism in centrosome [80]. To further dig into this mechanism, it would be interesting to investigate *spd-5*, as a main player. We overall concluded that it is plausible that a specific mechanism, distinct from the mammal one, ensures the nucleation of branched microtubules in the nematode zygote.

In a second example of using the predictive power of our three PCA coefficients, we expanded the investigation on *tpxl-1*, considering the debated mechanisms of the setting of the spindle length in the metaphasic zygote through 2 models. The first one by Greenan and co-authors proposed that, in the nematode, centrosome size, measured via the fluorescence of Spd-2 and Sas-4, correlates with spindle length [73]. The mechanism involves a Tpxl-1

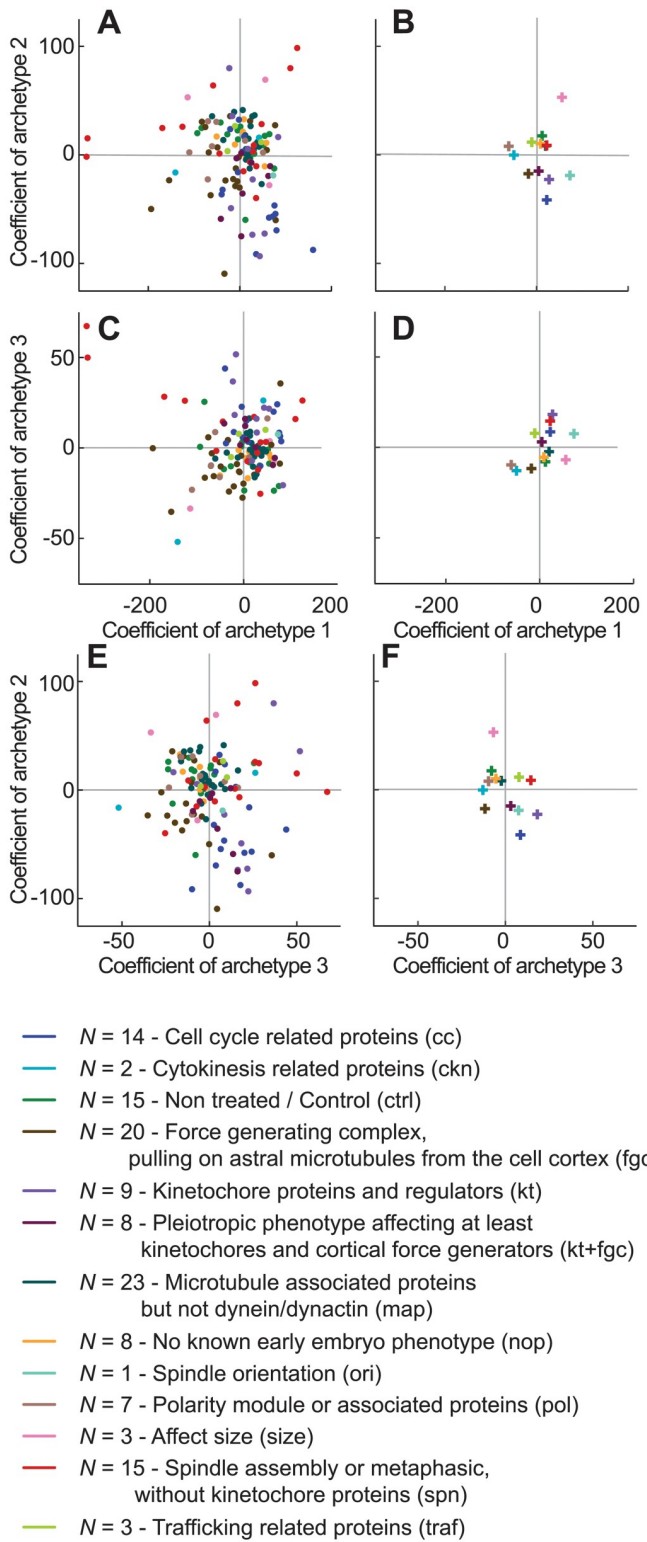

**Fig 6. PCA coefficients as a quantitative phenotype.** Each experiment is projected by PCA, and then a median is computed per condition. The resulting scatter plot is attached as an interactive plot (S1 File). (**A, C, E**) report on the coefficients 1 and 2, 2 and 3, 1 and 3 respectively. We colour-coded the conditions depending on their functional group as reported in (S1 Table). (**B, D, F**) depict the corresponding coefficients for each group computed as the median of the values per conditions plotted on the left-hand side panels. The corresponding spindle elongations are

reported at S7 Fig. The colour code for the group is shown in the bottom part, together with the abbreviated group within parentheses.

decreasing gradient from the centrosomes to the kinetochores and Air-1$^{\text{Aurora A}}$. In contrast, neither Tbg-1$^{\gamma\text{-tubulin}}$, Tac-1$^{\text{TACC}}$, Zyg-9$^{\text{XMAP215}}$ nor Klp-7$^{\text{MCAK}}$ were proposed to be involved. A second model in mammalian cells relates the spindle length to spindle microtubule poleward flux and involves Kif4A$^{\text{Klp-19}}$, Eg5$^{\text{Bmk-1}}$, Kif15$^{\text{Klp-18}}$, HSET somewhat close to Klp-15/16/17, NuMA$^{\text{Lin-5}}$ and MCAK$^{\text{Klp-7}}$ [81]. We repeated our predictive approach, providing *tpxl-1* and *air-1* as genes of interest to challenge the first model. We then intersected the set of genes predicted by the trained logistic regression (S8 Table) and found 9/14 predicted interactions not yet present in wormbase. Our approach predicted the expected *spd-2*, and consistently did not include *zyg-9* or *tac-1* as suggested by Greenan and co-authors. We did not find *sas-4* as it was not present in our dataset. Surprisingly, interaction prediction included *Klp-7*. This is the only homologous protein involved in the poleward flux mechanism in mammalian cells reported by Steblyanko and colleagues. Therefore, from our method point-of-view, the Greenan mechanism is more likely. To further test the Steblyanko flux-based mechanism, we nucleated our interaction predictor with the 8 genes involved in this mechanism as described above. After intersecting the interactor sets for these 8 genes, none of them remained in the resulting set (S9 Table). We next seeded our algorithm with only the 4 genes with pronounced phenotypes in the nematode. Indeed, *bmk-1* may diverge from classic sliding of overlapping microtubules [82] and *klp-15/16/17* are redundant, making their characterising complex. However, it resulted in the same result as with the 8 genes as seeds. Failure to find the initially seeded genes in the intersection of interactors suggested that they are unlikely to contribute to the same mechanism.

As a third example, we focused on anaphase spindle length, which classically involves the central spindle proteins Zen-4$^{\text{Mklp1}}$ and Spd-1$^{\text{Prc-1}}$ [83]. We seeded in with a single gene, *spd-1*. We predicted interactors in our dataset as above (S10 Table) and found 23/25 predicted interactions not yet present in wormbase. We recalled *zen-4* [84]. We also found the expected *cls-2* [85] not in wormbase-predicted interactors of *spd-1*. Interestingly, we also found Klp-7$^{\text{Mcak}}$ and Klp-19$^{\text{Kif4a}}$ previously involved in anaphase spindle length regulation in drosophila [86]. In particular, we and others recently suggested that Klp-19 may regulate microtubule growth during anaphase, contrasting with its supposed role as chromokinesin [87, 88]. Compared to *drosophila*, we did not find Bmk-1$^{\text{Eg5}}$, maybe because of its weak phenotype in the nematode and its role as a brake on elongation [82]. Overall, it suggests that the PCA coefficients map can be used to enrich the list of putative functional interactors, contributing to saving time in disentangling the complex mechanisms regulating the spindle. It also suggests that combining several genes reduced the number of false positives.

## Discussion

The specific question of extracting quantitative archetypes of inter-individual variability has received interest from the cellular level [23, 35] to the individual level [44, 89, 90] in the C. *elegans* research community. Our work complements and departs from these studies in several ways. In their study, Farhadifar et al. aimed at linking phenotypic diversity in spindle elongation with mutation accumulation over several generations or with evolutionary divergences between various strains [23]. In contrast, we focus here on linking spindle elongation phenotypes with perturbations of key molecular players in the most standard C. *elegans* lab strains to shed light on the molecular mechanisms underlying variability in spindle elongation.

Importantly, we uncovered a feature (archetype) corresponding to a limited elongation up to shortening during late metaphase, present in all conditions, including the non-genetically perturbed ones, by letting the projection method extract the most suited descriptors of variability for our dataset.

Interestingly, using PCA on time series provides interpretable archetypes describing variability. Our first archetype is reminiscent of the spindle-length average and correlated well with this intuitively-set feature (S1 Text and Fig 2E) [23]. Interestingly, the average spindle length was a feature used in many studies interested in the diversity of spindle elongation [23, 34, 91–93]. Finding such a feature through a data-centred approach suggests that the initial setting of spindle length in early metaphase is highly determinative of its dynamics over time. This is true even across anaphase onset, as the above predictive assay showed. Our second archetype included the elongation at anaphase, also previously identified as important to classify spindle elongation phenotypes, although only the anaphasic elongation was considered [23, 34, 94, 95]. In contrast, our second archetype displays two different slopes (Fig 2A, orange curve), one for the late-metaphase elongation —roughly starting 100 seconds before the metaphase-anaphase transition —and one for the anaphase elongation —starting at metaphase-anaphase transition. Finding both these slopes in the same archetype is a hint that the mechanisms of spindle dynamics before and after anaphase onset share commonalities so that they are tightly correlated. Consistently, we found that spindle length before anaphase onset is a predictor of spindle length in late anaphase meanwhile considering only late metaphase offers predictions that reasonably correlate with actual final lengths (Fig 4). Indeed, the establishment of the central spindle, although not fully understood, requires a re-assembly from scratch of the microtubules [85, 94, 96–99]. While the central spindle only appears after anaphase onset, our approach suggests that this re-assembly depends on the metaphasic spindle, through some mechanisms yet to be uncovered. In conclusion, our approach aims to give a unified view of the mitosis across phases to highlight the interdependence of the phenotypes.

Our third archetype is definitely novel and discovered thanks to our approach without a priori. It corresponded to a transient spindle limited-elongation phenotype during late metaphase. A plain shortening has been described in particular conditions with defective kinetochore-microtubule attachment dynamics [47, 49, 50, 100, 101]. The corresponding conditions present in this study displayed a significantly increased third coefficient (Fig 6 and S3 Fig and S1 Text and S6 Table). Yet, it has never been considered as a widespread phenomenon, nor used in describing variability in spindle elongation. At first glance, it might indeed seem paradoxical to include such a spindle-shortening phenotype but was clearly suggested since this coefficient contributes to accounting for variance even for non-treated embryos and many non-kinetochore-related conditions (S5 Table and Fig 6, S1(B), S1(C), S1(E) and S3 Figs). Yet, this phenomenon *naturally* occurs in wild-type embryos on a smaller magnitude than in genetically perturbed ones. Interestingly, targeting genes *ima-3* and *ani-2*, reported as reducing embryo length [3], also produced extreme coefficients 3. We related that to improper metaphasic spindle length as embryo-width was not reduced proportionally to length in these depletions, and so did the cell volume, putatively misleading the spindle size regulation [93]. Finally, it is interesting to notice that *tpxl-1(RNAi)* turned out to have a high coefficient 3 together with *cls-2(RNAi)* despite it was rather primarily reported to regulating spindle length [73]. It is consistent with the recent suggestion of its role in regulating the SKA complex, which in turn locks the kinetochore-microtubule attachments in late metaphase [102]. In a broader take, investigating conditions whose coefficient 3 departs from control could shed light on late metaphase regulation of spindle length, likely related to correct chromosome attachments.

Beyond finding archetypes informative on the mechanisms governing spindle length, we found that only three archetypes, spanning metaphase and anaphase, recapitulate 95% of the variability. Interestingly, the two first components are subtly departing from the manual features, namely spindle length and elongation rate, albeit being correlated with it (Fig 2F). These feature changes, together with the third archetype that accounted for 6%, allowed us to account for 95% of cell-to-cell heterogeneity of spindle elongation with only three archetypes, compared to about 40% with the 20 manually-set features previously proposed [2]. The fact our data-centred feature combines several manual features (arrows in Fig 2E–2G are neither vertical nor horizontal) suggest in contrast a correlation between the manual features. Indeed, PCA produces independent components (archetypes). Consequently, PCA can also be used to query genes that have multiple roles, revealed by all coefficients significantly differing from non-genetically perturbed conditions in the same strain and temperature (S11 Table). The reduced number of features can also be understood as strong constraints on the biological system studied. As experimental data can be rebuilt only using these few archetypes, the range of possible experimental observations is bounded. This highlights that our approach not only quantitatively detects variability but also reveals robust aspects of spindle elongation. Our data-centred approach offers an integrative view across time (phases) and mechanisms usually not accessible through explicit modelling because of the complexity of the mechanisms and the large number of players.

In a broader take, it raises the fundamental question of defining what is a phenotype in our biological system. Indeed, our archetypes can be described as phenotypes for extreme embryos (Fig 2B–2D), complying with the usual definition of the term: observable characteristics. Previous papers have indeed reported shorter spindles and spindle weakening leading to fast elongation or spindle shortening [31, 34, 94, 103–105]. Interestingly, our work suggests that phenotypes can be represented as the linear combination of the three archetypes. From the cell biology point of view, it suggests that all mechanisms are present in the non-treated embryos, although some are mildly contributing to spindle length and could only be very visible upon perturbation. Therefore, it suggests that rather than describing an experiment with discrete and controlled vocabulary like phenotype ontology [46], that is, whether a phenotype is absent or present, it would be more accurate to adopt a continuous description, namely the relative weights of archetypes. Such quantitative phenotype appears to us highly desirable as it enables the use of predictive algorithms in support of experimental work.

Using a dimensionality reduction approach without a priori also enabled us to investigate the complex spindle dynamics and propose only three archetypes [31–33]. Beyond the spindle length, a similar approach could be used to investigate other phenotypic traits. The variability may likely be described by a small number of archetypes, in many cases, providing descriptors in reduced numbers compared to existing approaches. For instance, quantifying C. *elegans* shape and movements is difficult, given the large apparent diversity in a given population. While previous studies have undergone a meticulous study of variability between individual shapes and behaviours by selecting up to several hundreds of features, unsupervised projection methods have identified four archetypes which recapitulate 97% of the observed variability in the dataset [90, 106].

Finally, mapping conditions in the PCA plane (Fig 6) provides indications of similarity between conditions. Such similar displacements along one of the axes in the PCA map often indicate some commonalities in the underlying molecular mechanisms. As such, when investigating a spindle-related gene with little to no documented function, one could partially impair its expression, monitor spindle elongation and project the result in the PCA map. Therefore, our PCA analysis could turn into a prospective tool for finding gene candidates for a mechanism by similarity to known players. This approach could also be instrumental in comparing

various strains of C. *elegans* or even various nematode species. Indeed, distance in the PCA plane can be seen as a phenotypic distance between conditions [107–111], more quantitative than the ones using phenotype ontology [46]. Along that line, we mapped our different experiments using the proposed three PCA coefficients to determine closely related genes through their quantitative phenotypes (Fig 6). Using machine learning, namely logistic regression, we predicted novel interacting genes not yet present in databases but for which experiments exist to validate the prediction. While we don't exclude the presence of false positives in the output, we foresee that such a bioinformatics approach could identify candidate genes in incomplete mechanisms to be further investigated experimentally.

## Materials and methods

### Culturing *C. elegans*

This study partly re-used experiments, which we have previously published, as referenced in S1 Table. In all cases, *C. elegans* nematodes were cultured as described in [112] and dissected to obtain embryos. The strains were maintained and imaged at temperatures between 16˚C and 25˚C (S1 Table). The strains were handled on nematode growth medium (NGM) plates and fed with OP50 E. *coli* bacteria.

### Strains used in this study

Strains carrying mutations or fluorescent labels in use in this study are detailed in S2 Table. Some strains were obtained by crossing, as detailed in this table.

### Gene inactivation through protein depletion through RNAi by feeding

RNA interference (RNAi) experiments were performed by feeding using either Ahringer-Source-BioScience library [45] with the bacterial strains detailed in S1 Table, either bacteria transformed to express dsRNA targeting the desired gene. The bacterial clone (bact-16) targeting *par-4* is a kind gift from Anne Pacquelet and was, in turn, obtained from [113]. These latter sequences are either from Cenix bioscience and are documented in wormbase [46] or were designed in the framework of this project (S3 Table). JEP:vec-33 and JEP:vec-35 were designed against *ebp-1* and *ebp-1/3*, respectively, and were disclosed in [114]. JEP vectors were constructed using Gateway technology (Invitrogen). Most RNAi treatments were performed by feeding the worm with specific bacterial clones. Feeding plates were either obtained by growing a drop of bacteria mixed with IPTG at the centre of a standard NGM plate or laying bacteria on a feeding plate, i.e. an NGM plate with indicated IPTG concentration in the agar. In all cases, plates were incubated overnight [115]. Alternatively, RNAi were obtained by injection of dsRNA in the gonads after [116]. The dataset includes control embryos for the RNAi experiments, obtained by feeding with bacteria carrying the empty plasmid L4440. We did not notice any phenotype suggesting that the meiosis was impaired during these various treatments.

### Embryos preparation and imaging

Embryos were dissected in M9 buffer and mounted on a pad (2% w/v agarose, 0.6% w/v NaCl, 4% w/v sucrose) between a slide and a coverslip. Embryos were observed at the spindle plane using a Zeiss Axio Imager upright microscope (Zeiss, Oberkochen, Germany) modified for long-term time-lapse. First, extra anti-heat and ultraviolet filters were added to the mercury lamp light path. Secondly, to decrease the bleaching and obtain optimal excitation, we used an enhanced transmission 12-nm bandpass excitation filter centred on 485 nm (AHF

analysentechnik, Tübingen, Germany). We used a Plan Apochromat 100/1.45 NA (numerical aperture) oil objective. Images were acquired with an Andor iXon3 EMCCD (electron multiplying charge-coupled device) 512 × 512 camera (Andor, Belfast, Northern Ireland) at 33 frames per second and using Solis software. Embryos were imaged at various temperatures as reported in S1 Table. To confirm the absence of phototoxicity and photodamage, we checked for normal rates of subsequent divisions in our imaging conditions. Images were then stored using Omero software [119] and analysed from there [117, 118].

## Centrosome-tracking assay, filtering and getting spindle length

The tracking of labelled centrosomes and analysis of trajectories were performed by a custom tracking software developed using Matlab (The MathWorks) [60, 61]. In a nutshell, because the centrosome is large enough to span over about ten pixels in width, we can correlate its image with a template. We then threshold the peak in correlation and get the centre of mass as the centrosome position. In the benchmarks, the approach by correlation is robust to intensity and size variations [120]. To validate the accuracy, we had tracked -20ºC methanol-fixed γ-tubulin labelled embryos and analysed the position fluctuation through Fourier analysis, finding an accuracy of 10 nm [60]. Embryo orientations and centres were obtained by cross-correlation of embryo background cytoplasmic fluorescence with artificial binary images mimicking the embryo or by contour detection of the cytoplasmic membrane using background fluorescence with the help of an active contour algorithm [121]. In this work, we accurately monitor spindle elongation from about 120 seconds before anaphase onset and up to 120 seconds after, which corresponds to the most dynamic phases of spindle elongation.

To exclude the rare tracking outliers, i.e. time-points at which the centrosome was confused with a transient bright spot briefly appearing, we compared the raw data points to median-track computed by a running median over a 3 s window. Each point at a distance larger than 3 μm was excluded. It could correspond to a displacement at 1 μm s$^{-1}$ or faster, which is about one order of magnitude larger than the maximum speed observed for centrosome during elongation or when spindle breaks (targeting *cls-2* e.g.). We repeated this filtering procedure a second time, computing the median on the data points filtered in the first time. Finally, we applied quality control and ensured that no more than 1% of the points were removed along the whole trajectory. Embryos not complying with this condition were excluded from further analysis. They corresponded to acquisition issues like poor focus. In other embryos, the remaining raw data points after filtering are used for subsequent analysis. The spindle length is computed as the Euclidean distance between the two centrosomes.

**Dataset.**   Our dataset included spindle elongation sampled at typically 30 frames per second, from 1618 experiments covering 78 gene depletions or mutations and realised at temperatures ranging from 15˚C to 25˚C, referred to hereafter as the whole dataset (S1 Table). These experiments included non-treated conditions at various temperatures or labellings, and control conditions (L4440, Methods). To further increase diversity, our dataset also comprised protein depletions through hypomorphic or penetrant RNAi, or mutants of genes involved in spindle positioning and mechanics. In particular, we tested all kinesins, that is, the 19 ones with interpro *Kinesin motor domain* homology [122, 123], microtubule-binding proteins (29 over the 64 with *microtubule-binding* GO term [51, 124]. The experimental dataset also featured subunits *dli-1*, *dylt-1*, *dyci-1*, *dhc-1* from dynein complex and *dnc-1* from dynactin [125].

## Quantifying KLP-7 expression level by fluorescence

The expression level of KLP-7 was assessed by quantifying fluorescence using Image J software. From the equatorial section of the zygote, at the anaphase onset, the average fluorescence intensity of the cell was measured, and the average fluorescence outside the embryo (background) was subtracted.

## Supporting information

**S1 Fig. Selecting the number of components for PCA projection.** (**A**) Decrease of the L2 error, i.e. the sum of the squares of the residuals, when increasing the number of dimensions in the PCA method. We suggest that 3 dimensions are optimal as it corresponds to the corner of the L-shaped curve (arrowhead). (**B**) Percentage of explained variance by each PCA component. (**C, E**) (grey) Comparing the raw spindle elongation of an exemplar single embryo labelled by GFP::ɣTUB, treated by (C) *cls-2(RNAi)* during 24 h or (E) non treated, and imaged at 18˚C. (coloured curves) We reconstructed the variability around the average with 1–3 and 10 PCA components and added the average elongation of all conditions used in this paper. The third component (archetype) was essential to recapitulate the key features of the experimental trajectories, especially the transient spindle limited-elongation / shortening before anaphase onset. (**D**) Histogram of clustering scores from a PCA with scrambled labels (§1 in S1 Methods) to be compared to PCA with real labels scoring 1.86. The red line depicts the maximum likelihood Gaussian fit.
(TIF)

**S2 Fig. Spindle elongation in non-treated embryos imaged at 18˚C with low coefficient associated to archetype 3.** Green thin lines report individual embryo elongation curves for the $N = 5$ embryos with lowest coefficient 3, $\overline{C_3^-} \simeq -21.4 \pm 4.3$. We first selected embryos with coefficient 2 between the first and third quartiles, termed mid-coefficient 2 embryos ($N = 28$). Then among these, we took 15% of the embryos with extreme coefficient 3. Doing so, coefficient 3 of the extreme pool is clearly different compared to the one of mid-coefficient 2 embryos, $\overline{C_3} \simeq -5.55 \pm 2.05$. In contrast, the coefficients 1 and 2 are similar in the two groups; they read $\overline{C_1^-} \simeq 53.9 \pm 16.9$ and $\overline{C_2^-} \simeq 6.8 \pm 3.7$ for the group with lowest coefficient 3 compared to $\overline{C_1} \simeq 47.2 \pm 5.02$ and $\overline{C_2} \simeq 9.73 \pm 1.23$ for mid-coefficient 2 group. The thick coloured line corresponds to the averages over these groups. The thicker blue line corresponds to the average over mid-coefficient 2 embryos. Compared to this latter average, a faster spindle elongation in late metaphase is visible in the low-coefficient-3 embryos average. All experiments were done using strain TH27, acquired at 18˚C. Individual embryo and averaged tracks were smoothed using a 1.5 s-running-window median.
(TIF)

**S3 Fig. Projection of the spindle elongations of individual embryos from a few conditions, including the ones presenting extreme coefficients (Fig 2B–2D) and non-treated ones.** Imaging was performed at 18˚C except otherwise stated. (**A**) Coefficients corresponding to the first two main archetypes (PCA components). (**B**) Similar plots for the second and third archetypes, and (**C**) for the first and third archetypes. Colours refer to genetic perturbations. Grey lines depict the 0 on each axis. All experiments were done using strain TH27 except the ones featuring *air-2(or207)*, which used JEP31. An interactive 3D plot is attached as S2 File. (**D**) Pole-pole distance (spindle length) averaged per condition and plotted during metaphase and anaphase for the cases displayed in panels A-C. Multiple conditions treating the same gene by RNAi or mutating it are merged. Averaged tracks were smoothed using a

1.5 s-running-window median. The black thicker line corresponds to the average over the whole dataset.
(TIF)

**S4 Fig. Spindle elongation averaged over non-treated embryos imaged at (blue curve) 18˚C and (orange curve) 23˚C.** Black thicker line corresponds to the average over the whole dataset, including all conditions. All experiments were done using strain TH27. Tracks were smoothed using a 1.5 s-running-window median.
(TIF)

**S5 Fig. Clustering of conditions per function group.** We shuffled the group labels with respect to S1 Table and computed the Kruskal-Wallis $H$ to assess whether conditions from the same group clustered. (**A-C**) We repeated this computation 10000 times and reported the distribution of $H$ for each coefficient. The arrows indicate the values obtained with true labels for each coefficient, $H_1 = 17.9$, $H_2 = 28.7$ and $H_3 = 39.9$. Red lines depict the maximum-likelihood Gaussian fit.
(TIF)

**S6 Fig. Archetypes upon PCA of kinetochore functional group embryos.** (**A**) Averages of the three first PCA archetypes computed considering only the embryos from conditions of the group "kinetochore proteins and regulators" (kt) ($N$=86) and compared to (dashed lines) archetypes extracted from the whole set of conditions ($N$=1618). The elongation curves were smoothed with a 1.5 s running-median filtering before computing PCA. Explained variance is reported in S5 Table. (**B**) The corresponding spindle elongation was computed as the median of the average elongation curve among embryos from the same conditions. The track was smoothed using a 1.5 s-running-window median. The black thicker line corresponds to the average over the whole dataset, including all conditions.
(TIF)

**S7 Fig. Spindle elongation per group computed as the median of the curves obtained for each condition in the group.** In turn, the elongation for each condition is computed as the average of the curves for each embryo within the condition. Each group track was smoothed using a 1.5 s-running-window median. The black thicker line corresponds to the average over the whole dataset, including all conditions. The corresponding PCA values are reported at Fig 6.
(TIF)

**S8 Fig. Spindle elongation trajectories depend on the penetrance of RNAi.** Exemplified through RNAi targeting KLP-7[MCAK] in KLP-7::mNG background (strain LP447) in three conditions: (**A**) $N$=11 non-treated embryos; (**B**) $N$=18 *klp-7(RNAi))* treated embryos; (**C**) $N$=8 control embryos (L4440 treated). The thick lines report the averages of each condition and correspond to the data in Fig 5A. The three conditions reported here were not included in the initial dataset used to generate PCA archetypes. Acquisitions were performed at 18˚C. The line colour encodes the fluorescence level of KLP-7::mNG (Methods).
(TIF)

**S1 File. PCA coefficients averaged per condition and interactively plotted in 3D.**
(HTML)

**S2 File. PCA coefficients for individual embryos interactively plotted in 3D for conditions plotted in S3 Fig.**
(HTML)

**S1 Text.** (1) Multilinear fit predicting Farhadifar features by regression on the PCA coefficients and (2) genes causing spindle shortening or limited elongation in late metaphase.
(PDF)

**S1 Methods. Supplemental methods.**
(PDF)

**S1 Table. Treatments used in this study.** The fluorescent strains used, carrying possibly mutated genes, are detailed in S2 Table. Target identifies the condition in the figures. RNAi treatments were performed by feeding or injection as detailed in the methods. L4440 corresponds to control conditions for RNAi treatments. Group corresponds to manual tags by function in the one-cell embryo based on papers and known phenotypes in wormbase [79]. Group abbreviations are detailed in Fig 6.
(PDF)

**S2 Table. Fluorescently tagged strains used in this study and their detailed genotypes.** Original strains are referenced by each of the crossed strains, whereas previously disclosed ones are referenced by the corresponding publication.
(PDF)

**S3 Table. Bacterial clones designed for this study to silence genes by RNAi.**
(PDF)

**S4 Table. Ratio of inter-group variance over intra-group variance.** We compared various projection methods by assessing their ability to cluster replicas while separating experiments corresponding to distinct treatments, using the score described in S1 Methods. We also included some non-linear/local methods for the sake of completeness, although they will not enable the interpretability expected in our specifications. Higher scores mean that the projection method performs better. The last row correspond to the average upon 10000 repeats of shuffling experiment-labels and computing the score.
(PDF)

**S5 Table. Explained variance for PCA over a subset of dataset.** We performed a PCA analysis on a subset of the dataset and obtained the reported percentage of explained variances (see details in main text and Fig 3). N corresponds to the number of embryos in each set.
(PDF)

**S6 Table. Mann-Whitney test comparing coefficients of genes for which depletion was previously reported as causing spindle shortening during late metaphase.** The listed treatments performed at 18˚C were achieved by RNAi on the TH27 strain and compared to the corresponding L4440 treated embryos. In contrast, *air-2* was a mutant reported as temperature sensitive, although we already observed phenotype at permissive temperature. It was compared to non-treated embryos from the TH27 strain at the closest temperature. Distributions are represented at S3 Fig.
(PDF)

**S7 Table. Predicting genes interacting with *tpxl-1*.** We trained a logistic regression with wormbase-known and -predicted interactors (column known interaction to true) among tested proteins in our dataset and predicted additional interactors marked as True in column predicted interaction (§5 in S1 Methods).
(PDF)

**S8 Table. Predicting genes interacting with *tpxl-1* and *air-1*.** We trained a logistic regression with wormbase-known and -predicted interactors (column known interaction to true) among tested proteins in our dataset and predicted additional interactors marked as True in column predicted interaction (§5 in S1 Methods).
(PDF)

**S9 Table. Predicting genes interacting with *klp-19*[kif4A], *bmk-1*[eg5], *klp-18*[kif15], *klp-15/16/17* somewhat close to hset, *lin-5*[numa] and *klp-7*[mcak].** We trained a logistic regression with wormbase-known and -predicted interactors (column known interaction to true) among tested proteins in our dataset and predicted additional interactors marked as True in column predicted interaction (§5 in S1 Methods).
(PDF)

**S10 Table. Predicting genes interacting with *spd-1*.** We trained a logistic regression with wormbase-known and -predicted interactors (column known interaction to true) among tested proteins in our dataset and predicted additional interactors marked as True in column predicted interaction (§5 in S1 Methods).
(PDF)

**S11 Table. Conditions where all three coefficients depart from control.** Mann-Whitney test comparing coefficients of genes for which a significative difference ($p < 0.01$) was found for all coefficients. The listed treatments were achieved by RNAi on the TH27 strain and compared to the L4440-treated control embryos at the same temperature.
(PDF)

## Acknowledgments

Strains TH27, TH231, TH290, TH291, TH243 were a kind gift from Prof Anthony A. Hyman. Some strains were provided by the Caenorhabditis Genetics Center (CGC), which is funded by National Institutes of Health Office of Research Infrastructure Programs (P40 OD010440; University of Minnesota). Strain ANA019 was kindly offered by Dr Marie Delattre. Some strains were provided by NBRP, which is funded by the Japanese government. The bact-16 bacteria to perform *par-4(RNAi)* is a kind gift from Dr Anne Pacquelet. We thank Dr. Gregoire Michaux for the feeding clone library and technical support. We also thank Drs. Grégoire Michaux, Anne Pacquelet, Sébastien Huet, Marc Tramier and Olivier Dameron for discussions about the project. Microscopy imaging was performed at the Microscopy Rennes Imaging Center, UMS 3480 CNRS/US 18 INSERM/University of Rennes.

## Author Contributions

**Conceptualization:** Yann Le Cunff.

**Data curation:** Laurent Chesneau, Sylvain Pastezeur, Xavier Pinson, Nina Soler, Danielle Fairbrass, Benjamin Mercat, Ruddi Rodriguez-Garcia, Zahraa Alayan, Ahmed Abdouni, Gary de Neidhardt, Valentin Costes, Mélodie Anjubault, Hélène Bouvrais, Jacques Pécréaux.

**Formal analysis:** Yann Le Cunff, Jacques Pécréaux.

**Funding acquisition:** Yann Le Cunff, Hélène Bouvrais, Jacques Pécréaux.

**Investigation:** Yann Le Cunff, Laurent Chesneau, Jacques Pécréaux.

**Methodology:** Yann Le Cunff, Jacques Pécréaux.

**Project administration:** Yann Le Cunff, Jacques Pécréaux.

**Software:** Yann Le Cunff, Hélène Bouvrais, Jacques Pécréaux.

**Supervision:** Jacques Pécréaux.

**Validation:** Yann Le Cunff, Hélène Bouvrais, Christophe Héligon, Jacques Pécréaux.

**Visualization:** Yann Le Cunff, Jacques Pécréaux.

**Writing – original draft:** Yann Le Cunff, Jacques Pécréaux.

**Writing – review & editing:** Yann Le Cunff, Hélène Bouvrais, Christophe Héligon, Jacques Pécréaux.

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
