## [Decision Letter · Decision Letter 0]

2 Apr 2024

Dear Dr Pécréaux,

Thank you very much for submitting your manuscript "Unveiling inter-embryo variability in spindle length over time: towards quantitative phenotype analysis." for consideration at PLOS Computational Biology.

As with all papers reviewed by the journal, your manuscript was reviewed by members of the editorial board and by several independent reviewers. In light of the reviews (below this email), we would like to invite the resubmission of a significantly-revised version that takes into account the reviewers' comments.

We cannot make any decision about publication until we have seen the revised manuscript and your response to the reviewers' comments. Your revised manuscript is also likely to be sent to reviewers for further evaluation.

Sincerely,

Changbong Hyeon

Academic Editor

PLOS Computational Biology

Jason Haugh

Section Editor

PLOS Computational Biology

Reviewer's Responses to Questions

**Comments to the Authors:**

Reviewer #1: The authors of this interesting and useful paper approach the problem of

dynamic variability of mitotic spindle using data science approach, without

fitting or modeling or pre-determining quantitative features of the spindle.

They take measured time series of spindle lengths in metaphase and anaphase

under many control and perturbed conditions, and apply PCA analysis to these

time series. Most of variability is captured by just three first eigenvectors,

which the authors call archetypes. Those archetypes have intuitive interpretation:

one of them characterizes the average spindle length, another - the elongation

rate, and the third one - transient 'weakening' of the spindle in late metaphase.

By choosing various subsets of the data, the authors demonstrate that these

archetypes are robust. They use ML to show that weights of the archetypes at the

anaphase onset predict the anaphase elongation rate.

The paper is sound and timely. I have the following three general comments:

1) It is a bit unsatisfying that there is little insight into molecular mechanisms

of mitosis. After all, many specific perturbations are used. I sense that by

looking at the weights of the archetypes in different conditions we can learn more

about plausibility of many mechanistic models of the spindle dynamics proposed in the

last decade.

2) Aside from criticism of studies [2,23], the features pre-selected there were

very useful for classifying mitotic scenarios. Are there any quantitative correlations

between the weights of the archetypes and these previously introduced signatures of the

spindle elongation. Such analysis would be enlightening.

3) Writing is mostly clear, but overly 'wordy'. I felt, when reading, that the main

text could be significantly shortened - there are many repetitions and meandering

descriptions. Also, the section on using ML to show that weights of the archetypes at

the anaphase onset predict the anaphase elongation rate probably belongs to the end of

the results - it breaks the flow, and the robustness results have to be shown before it.

Reviewer #2: The manuscript by Le Cunff et al. tackles a long-standing issue in modeling and understanding the in vivo behavior of biological structures – the variability that occurs among different measurements of the same parameter. The authors utilized PCA instead of a more traditional mathematical model approach with the aim of better accounting for variability in raw data. PCA analysis allowed the authors to “extract” different types of spindle elongation behaviors from multiple experimental manipulations of the first mitosis in C. elegans embryos and demonstrate that these “archetypes” are present in unperturbed embryos. This work provides solid support for relying less on fitting models to data based on a priori knowledge and instead allowing unbiased methods to highlight features that may not be appreciated or identified when using a pre-existing framework. This proof of principle may be useful to multiple fields of biology.

From a cell biologist’s perspective, it would be beneficial to dive a little deeper into how this approach our understanding of spindle elongation. While the diverse range of spindle-associated proteins subjected to partial RNAi is commendable, much of the analysis was primarily focused on seeing how the archetypes derived from PCA were affected by different subsets of the experimental data. Further analysis of specific RNAi conditions that were not highlighted in the main text may yield new insights on characterized spindle proteins being involved in more pathways than previously known. Furthermore, the authors propose that the RNAi and PCA approach shown here may help to identify novel spindle proteins that were previously elusive due to hard-to-interpret phenotypes; a concrete example of this case (i.e. showing a protein depletion that the PCA matches to a specific archetype and verifying a new function for that protein) would significantly boost the impact of the work. Nevertheless, the general premise of the work seems sound and logical; the experiments and computations appear rigorous and well-designed and this work seems suitable for publication in PLOS Computational Biology.

Major points

1. A stronger emphasis on how PCA differs from previous mathematical a priori modeling of spindle dynamics would be helpful. The motivation for using PCA instead of relying on well-established models in the field of spindle dynamics could be better described in the introduction. What exactly is gained by this approach? Additional explanation of how a priori mathematical models have not fully recapitulated spindle behavior would be beneficial to demonstrating the value for an approach that can find “hidden” features.

2. A definitive application of this approach to identify a novel spindle-related function of a previously ambiguous protein would be extremely convincing of the future application of this PCA approach.

Minor points

1. The provided supplemental data and tables (e.g. Table S6) highlight that certain RNAi conditions contribute strongly to one archetype. It would be intriguing if any of the RNAi conditions tested in the dataset have an almost equal contribution across all three archetypes, suggesting roles in multiple aspects of the mitotic spindle that may not have been previously suggested/identified. There are multiple proteins listed in the supplement (i.e. kinesins, TAC-1, ZYG-9) that might be predicted to have multiple effects on spindle elongation.

2. This study and cited papers (55,56), use computational calculations of centrosome position, how does this lead to “10 nm accuracy”, beyond the resolution limit of light microscopy?

3. Figure 5A is visually difficult to interpret. Having so many lines plotted together across a gradient of color and textures makes it difficult to discern which line corresponds to what condition and KLP-7 intensity.

4. Throughout the manuscript requires editing for clarity/accuracy.

**Have the authors made all data and (if applicable) computational code underlying the findings in their manuscript fully available?**

Reviewer #1: Yes

Reviewer #2: Yes

PLOS authors have the option to publish the peer review history of their article (what does this mean?). If published, this will include your full peer review and any attached files.

Reviewer #1: No

Reviewer #2: No
---

## [Decision Letter · Decision Letter 1]

15 Jul 2024

Dear Dr Pécréaux,

We are pleased to inform you that your manuscript 'Unveiling inter-embryo variability in spindle length over time: towards quantitative phenotype analysis.' has been provisionally accepted for publication in PLOS Computational Biology.

Best regards,

Changbong Hyeon

Academic Editor

PLOS Computational Biology

Jason Haugh

Section Editor

PLOS Computational Biology

Reviewer's Responses to Questions

**Comments to the Authors:**

Reviewer #1: I am satisfied with the revisions

Reviewer #2: The authors have addressed all of our concerns.

**Have the authors made all data and (if applicable) computational code underlying the findings in their manuscript fully available?**

Reviewer #1: Yes

Reviewer #2: Yes

PLOS authors have the option to publish the peer review history of their article (what does this mean?). If published, this will include your full peer review and any attached files.

Reviewer #1: No

Reviewer #2: No

---

## [Editor Report · Acceptance letter]

13 Aug 2024

PCOMPBIOL-D-24-00330R1 

Unveiling inter-embryo variability in spindle length over time: towards quantitative phenotype analysis.

Dear Dr Pécréaux,

I am pleased to inform you that your manuscript has been formally accepted for publication in PLOS Computational Biology. Your manuscript is now with our production department and you will be notified of the publication date in due course.

With kind regards,

Dorothy Lannert
